# Malaria awareness of adults in high, moderate and low transmission settings: A cross-sectional study in rural East Nusa Tenggara Province, Indonesia

**Robertus Dole Guntur**[1,2]*, **Jonathan Kingsley**[3,4], **Fakir M. Amirul Islam**[1]

**1** Department of Health Science and Biostatistics, Swinburne University of Technology, Hawthorn, Victoria, Australia, **2** Department of Mathematics, Faculty of Science and Engineering, Nusa Cendana University, Kupang, NTT, Indonesia, **3** Department of Health and Medical Science, Swinburne University of Technology, Hawthorn, Victoria, Australia, **4** Centre of Urban Transitions, Swinburne University of Technology, Hawthorn, Victoria, Australia

* rguntur@swin.edu.au

## Abstract

### Introduction

The 2009 Indonesian roadmap to malaria elimination indicated that the nation had been progressing towards achieving malaria elimination by 2030. Currently, most of the districts in the western part of Indonesia have eliminated malaria; however, none of the districts in the East Nusa Tenggara Province (ENTP) have met these set targets. This study aimed to investigate the status of malaria awareness of rural adults in the ENTP.

### Methods

A community-based cross-sectional study was conducted between October and December 2019 in high, moderate, and low malaria-endemic settings (MESs) in the ENTP. After obtaining informed consent, data were collected using an interviewer-administered structure questionnaire among 1503 participants recruited by a multi-stage cluster sampling method. A malaria awareness index was developed based on ten questions. A binary logistic regression method was applied to investigate the significance of any association between malaria awareness and the different MESs.

### Results

The participation rate of the study was 99.5%. Of this number, 51.4% were female and 45.5% had completed primary education. The malaria awareness index was significantly low (48.8%, 95% confidence interval [CI]: 45.2–52.4). Malaria awareness of rural adults residing in low endemic settings was two times higher than for those living in high endemic settings (adjusted odds ratio [AOR]: 2.41, 95% CI: 1.81–3.21) and the basic malaria knowledge of participants living in low malaria-endemic settings was almost four times higher than that in high endemic settings (AOR: 3.75, 95% CI: 2.75–5.11). Of the total participants,

**Funding:** PhD scholarship for RDG was supported by the Australia Awards Scholarship (ST000TBK6). The Faculty of Health, Arts and Design (FHAD) of the Swinburne University Technology supported for the primary data collection. The funders had no role in the design of the study, data collection, analysis, or interpretation of data or writing the manuscripts.

**Competing interests:** The authors have declared that no competing interests exist.

81.3% (95% CI: 79.1–83.5) were aware that malaria could be prevented and 75.1% (95% CI: 72.6–77.6) knew at least one prevention measure. Overall, the awareness of fever as the main symptom of malaria, mosquito bites as the transmission mode of malaria, and seeking treatment within 24 hours of suffering from malaria was poor at 37.9% (95% CI: 33.9–41.9), 59.1% (95% CI: 55.9–62.3), and 46.0% (95% CI: 42.3–49.7), respectively. The poor level of awareness was significantly different amongst the three MESs, with the lowest levels of awareness in the high endemic setting.

## Conclusion

Malaria awareness of rural adults needs to be improved to address Indonesia's national roadmap for malaria elimination. Results indicated that public health programs at a local government level should incorporate the malaria awareness index in their key strategic intervention to address malaria awareness.

## Introduction

Malaria is a major global health problem with an estimated 1.2 billion people living at a high risk of being infected [1]. However, malaria cases and associated deaths have decreased in the last decade: from 2010 to 2018, the total number of malaria cases decreased by approximately 1% per year and deaths due to malaria declined by 5% annually [1]. The number of countries reporting less than 51 cases of local transmission increased from 5 countries in 2010 to 11 countries in 2018 [2]. Countries with zero local transmission in the last three consecutive years are eligible to request malaria elimination certification from the World Health Organization (WHO) [2]. Two countries, the Maldives and Sri Lanka, have been certified as malaria-free areas by the WHO Regional Office for South-East Asia (SEARO) [3]. In alignment with the global action plan for a malaria-free world [4] and the Global Technical Strategy for Malaria Elimination [5], the WHO SEARO action plan indicates that all countries in the region will be malaria-free zones by 2030 [3].

The roadmap for malaria elimination in Indonesia was proposed in April 2009 and aimed to eliminate malaria by 2030 [6, 7]. All malaria-endemic districts in Indonesia were divided into four categories based on the annual prevalence incidence (API). Of the 514 districts in the country, 298 (58%) were categorized as malaria elimination districts in 2019 [8]. All districts in the provinces of the Special Capital Region of Jakarta, Bali and East Java have been categorized as malaria elimination areas, whereas none of the districts from five provinces in the eastern part of Indonesia such as Papua, West Papua, Maluku, North Maluku, and the East Nusa Tenggara Province (ENTP) have achieved this categorization [8].

The ENTP is a province with an API value that is five times higher than the national Indonesian level [9]. This province has 21 districts and one municipality [10]. Fourteen districts and the municipality are low endemic, while four and three districts have been classified as moderate and high endemic, respectively [8]. In line with the national commitment to eliminate malaria by 2030, there have been various efforts of the local authorities to support malaria elimination in this province. This has included increasing the coverage of artemisinin-based combination therapy (ACT) as the first line of malaria treatment from 55% in 2013 [11] to 83.1% in 2018 [9], and screening pregnant women for malaria during their first visit to local health centres [12]. For controlling mosquitoes, the introduction of treated bed nets has been

implemented in most of the districts in the region since 2008 [13], the mass distribution of long-lasting insecticide–treated nets (LLINs) in 15 districts since 2017 [14], and the use of special repellent [15]. However, the number of malaria cases is still high (12,909 cases) [8], indicating that these interventions may be ineffective and that the implementation of these interventions may depend on community behavior; however, there has been limited investigation of these factors in this province. Community knowledge and behavior play significant roles in supporting malaria elimination [16, 17]: high levels of malaria awareness in communities enables them to improve self-protection [18], seek early treatment [19], and reduce malaria prevalence [20], consequently speeding up malaria elimination [21].

Several studies concerning malaria knowledge have been undertaken in Indonesia since the declaration of the national commitment to eliminate malaria [22–25]. However, knowledge of LLINs, which are the most effective tools to prevent malaria [26] and are currently adopted as the primary vector control intervention in many parts of Indonesia [7], was not investigated in these studies. Additionally, most of the studies were conducted in the western part of Indonesia, which has been classified as a malaria elimination area. Studies were also conducted at the sub-district and village levels. One population-based study on 4,050 participants in North Maluku province revealed that only about half of the respondents knew about symptoms of malaria and the majority of participants (98%) did not know the main cause of malaria [24]. However, approximately 50% of the participants were less than 18 years old and were hardly suitable candidates for measuring the level of knowledge of a particular community.

Various studies on malaria knowledge have been conducted in the ENTP [12, 27–29]. Most of these studies were conducted at the village and subdistrict levels. One population study covering only pregnant women in a high malaria-endemic area of the province indicated that there was a low level of malaria prevention knowledge, particularly relating to LLINs [12]. Another population-level study on community behaviour relating to malaria was conducted in the ENTP in 2018 [29]. However, this study only investigated malaria prevention practices of the rural community and malaria prevention awareness in different MESs was not compared. To date, the investigation of malaria knowledge relating to the symptoms, transmission mode, and prevention method, and the malaria treatment-seeking behaviour of rural adults in different types of malaria-endemic settings in the ENTP has not been performed. Investigation of malaria awareness in rural communities is critical for Indonesia, considering that 52% of malaria cases in the country were contributed by rural communities [9] and that there are variations in malaria prevention practice amongst provinces in the country [29]. An understanding of the level of malaria knowledge in rural communities and determining which MES is most vulnerable is essential for the development and implementation of evidence-based strategies to accelerate progress towards malaria elimination in the province. The present study aimed to fill this gap by investigating malaria awareness of rural adults in three different MESs to support the national commitment of Indonesia's government to eliminate malaria by 2030.

## Materials and methods

### Study sites

The ENTP is one of 34 provinces in Indonesia, in the eastern part of the country. The total population of the ENTP is 5.3 million, accounting for about 2.04% of the total population of Indonesia [10]. The ratio of male to female (50.5% to 49.5%) is comparable with that of Indonesia (50.2% to 49.8%). The area of the province is 47,931.54 km$^2$, located between 1180˚ and 1250˚ east longitudes and between 80˚ and 120˚ south latitudes, with a population density of 114 people per square kilometer [10]. This community-based cross-sectional study was conducted from October to December 2019 in three districts out of 21 districts and one

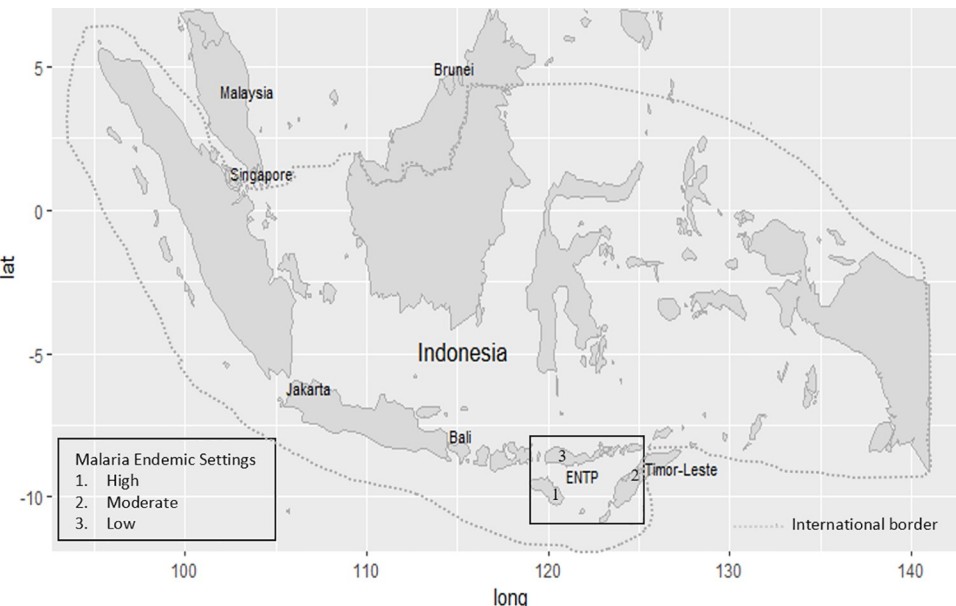

**Fig 1. Map of study sites.**

municipality in the province. They were East Sumba, Belu, and East Manggarai districts representing high, moderate, and low MESs, respectively [30], as shown in Fig 1.

## Sample size calculation

The initial sample size ($n_0$) was calculated based on the formula $n_0 = Z^2P(1-P)/d^2$ for the prevalence study of a cross-sectional study [31]. The parameters Z is the value of standard score of 95% confidence interval (1.96), P is the prevalence of malaria study in the ENTP conducted by the Indonesian government (1.99%) [9], and d is the relative precision which is 0.01125. Therefore, the initial sample size $n_0$ was equal to 592. The design effect was accounted for due to cluster sampling by the multiplication of a factor of 2.16. By considering the participation rate of 85%, the final sample size was 1503 adults. The sample size calculation was described previously [32].

## Sampling technique

All adults in ENTP were the source population, and all adults in the selected three districts were the study population. A multi-stage cluster sampling procedure with a systematic random sampling procedure at the final cluster level 4 was applied to recruit adults from the three districts. At cluster level 1, three districts were selected out of 22 in the ENTP based on the annual parasite incidence (API) of malaria, at cluster level 2, three sub-districts were randomly chosen from each selected district. At cluster level 3, the number of villages selected from each sub-district was based on their relative populations. At the final cluster level, a systematic random sampling technique was used to recruit 20–40 participants per village, proportionate to the population size of each village. In each selected household, one head of the family of any gender who provided consent to participate voluntarily was included in the study. If the household head, either husband or wife, was absent, residents over 18 years of age could serve as study participants [33]. We excluded anyone under the age of 18 years old from the study.

## Data collection tools and techniques

An interviewer-administered questionnaire adapted from a validated questionnaire [34, 35] was used to collect data for this study. The English version of the questionnaire was translated into the local language by the lead author of this article and a local language expert. They then combined the two translated versions. The combined version of the questionnaire was then tested on 30 participants before finalization. The data were collected in collaboration with local nurses who were residents in the study area. Nine local nurses, three nurses from each district conducted face-to-face interviews with participants based on the guidance of the structured questionnaire. The data collection process was monitored strictly by the investigator daily to check the questionnaire's completeness. Data on the socio-demographic variables and general knowledge of malaria was collected during the interview.

## Malaria awareness measures

Ten questions were used to assess the malaria understanding and knowledge of rural adults. The first three questions explored participants' basic understanding of malaria, including whether they had heard of malaria, whether malaria was dangerous to their health, and whether malaria could be prevented. The responses options for these questions were yes or no, with yes receiving a score of one. Overall, participants obtained a score of three if they correctly answered all three questions. The total score for participants' basic understanding of malaria was evaluated following the approach in previous studies [36, 37]. Participants who correctly answered at least two of the first three questions were categorized as having basic malaria understanding; participants were otherwise categorized as having no basic malaria understanding.

The next seven questions also explored basic malaria knowledge, including whether participants could identify: 1) the main symptom and cause of malaria, 2) protective measures to prevent malaria, and 3) the importance of seeking treatment for malaria within 24 hours after the onset of the symptoms. Participants who could identify fever as the main symptom [38] and mosquito bites as the main cause of malaria [39, 40] obtained a score of one for each. Participants who mentioned sleeping under non-LLINs, sleeping under LLINs, using mosquito coils, or keeping the house clean as methods to prevent malaria also achieved a score of one for each. Finally, participants who mentioned seeking malaria treatment within 24 hours [41] obtained a score of one. A total score of seven was possible if participants correctly answered all seven questions. The total score for participants' basic malaria knowledge was further evaluated following the procedure described in previous studies [36, 37]. Participants who correctly answered at least five of these seven questions were categorized as having basic malaria knowledge.

Overall, each participant could gain a score of ten if they correctly answered all ten questions. The total score for the ten questions was evaluated following the approach described in previous studies [36, 37]. Participants with scores of above 80%, 60–79%, 1–59%, and 0 were classified as having excellent, good, poor, and zero malaria knowledge, respectively; participants in the excellent and good groups were categorized as having malaria awareness while those who were in the poor and zero groups were classified as being unaware of malaria [36, 37].

## Socio-demographic covariates

Socio-demographic information including gender, age, education level, and socioeconomic status (SES) was collected. Gender was categorized as male and female. Age was classified into five groups, < 30, 30–40, 40–50, 50–60, and > 60 years old. The level of education was categorized as no education, primary school (grade 1 to 6), junior high school (grade 7 to 9), senior

high school (grade 10 to 12), and diploma or above. The SES group was assessed according to ownership of durable assets and housing characteristics [42]. In each selected household, the participant was asked about their ownership of ten durable asset items including radio, television, electricity, bike, motorcycle, handphone, fridge, tractor, generator, and car. Housing characteristics were evaluated according to access to water taps in dwellings and the main material of the house, with houses having cement floors and walls categorized as modern houses and others as non-modern houses. In total, 12 items were used to construct the SES level and three SES levels were defined by counting the overall ownership of these items. Low SES was defined as having zero or one item; moderate SES was defined as owning two to four items and high SES was defined as having more than four items, following the approach of Zafar et al. [42].

## Statistical analyses

The participants' socio-demographic characteristics including gender, age group, education level, and SES were reported using descriptive statistics. The proportion of participants answering each question correctly and its 95% confidence interval (CI) were computed for each MES. The association between the MES and responses to the 10 questions was explored by the chi-square method. This approach was further applied to initially evaluate the association of basic malaria understanding, basic malaria knowledge, the level of malaria knowledge, and the level of malaria awareness amongst the three types of MESs. A univariate and multivariate binary logistic regression model was applied to evaluate the association between the dependent and the independent variables. The associations were reported as odds ratio with its 95% CI. Multicollinearity tests amongst the independent variables were done before multivariate analysis was conducted. The Hosman and Lemeshow test evaluated the overall model fitness with a significance level of $p < 0.05$. Wald statistic was used to assess the significance of the individual covariates in the model. In the univariate binary logistic regression, all variables having a p-value $< 0.10$ were included in the multivariate analysis to control confounding factors [43]. After controlling the confounding variables, all variables with a p-value of 0.05 or less were considered statistically significant as a predictor of outcomes variables. The direction and strength of association between explanatory variables and endpoints were estimated by adjusting the odds ratio. Statistical software SPSS version 27 (SPSS Inc.) was used for analyses.

## Ethics approval

The research was conducted in accordance with the tenets of The Declaration of Helsinki. The study was approved by the Human Ethics Committee of Swinburne University of Technology, Australia (Reference: 20191428–1490) and the Health Research Ethics Committee, National Institute of Health Research and Development (HERC-NIHRD), Ministry of Health of Indonesia (Reference: LB.02.01/2/KE.418/2019). Written consent was obtained from participants who had the full capacity to give voluntary consent in their own right based on the provision of sufficient information. Participants who were unable to read the consent documentation authorized their spouse or immediate family member to read the consent form and sign it on their behalf. Participants were informed of their right to withdraw from the study at any stage or to restrict the use of their data in the analysis.

## Results

### Demographic characteristics of the study population

The participation rate of this study was 99.5% (1495 out of 1503). Of all the participants aged between 18 and 89 years (mean: 43.8 years, standard deviation: 12.8 years), 51.4% was female.

**Table 1. Distribution of study participants and participants from a national representative sample in three different MES in the East Nusa Tenggara Province (ENTP), Indonesia.**

| Characteristic | ENTP | Total n (%) | Malaria Endemic Setting (MES)[b] | | |
| --- | --- | --- | --- | --- | --- |
| | | | High | Moderate | Low |
| **Total** | 5,456,203 | 1,495 | 495 (33.1) | 500 (33.4) | 500 (33.4) |
| **Gender** [10] | | | | | |
| Females | 50.5 | 768 (51.4) | 264 (53.3) | 267 (53.4) | 237 (47.4) |
| Males | 49.5 | 727 (48.6) | 231 (46.7) | 233 (46.6) | 263 (52.6) |
| [a]**Age Group** [10] | | | | | |
| < 30 | 39.9 | 205 (13.7) | 79 (16.0) | 64 (12.8) | 62 (12.4) |
| 30–39 | 18.9 | 418 (28.0) | 137 (27.7) | 108 (21.6) | 173 (34.6) |
| 40–49 | 16.4 | 371 (24.8) | 138 (27.9) | 123 (24.6) | 110 (22.0) |
| 50–59 | 12.7 | 295 (19.7) | 69 (13.9) | 129 (25.8) | 97 (19.4) |
| > 60 | 12.1 | 206 (13.8) | 72 (14.5) | 76 (15.2) | 58 (11.6) |
| **Education Level** [10] | | | | | |
| No education | 30.4 | 279 (18.7) | 173 (35.0) | 93 (18.6) | 13 (2.60) |
| Primary school | 27.5 | 678 (45.4) | 205 (41.4) | 205 (41.0) | 268 (53.6) |
| Junior High school | 16 | 229 (15.3) | 47 (9.50) | 97 (19.4) | 85 (17.0) |
| Senior High school | 18.6 | 210 (14.1) | 53 (10.7) | 83 (16.6) | 74 (14.8) |
| Diploma or above | 7.6 | 99 (6.60) | 17 (3.40) | 22 (4.40) | 60 (12.0) |
| **Socio-Economic Status** [44] | | | | | |
| Poor | 89.6 | 449 (30.0) | 151 (30.5) | 105 (21.0) | 193 (38.6) |
| Average | 4.80 | 860 (57.5) | 286 (57.8) | 331 (66.2) | 243 (48.6) |
| Rich | 5.70 | 186 (12.4) | 58 (11.7) | 64 (12.8) | 64 (12.8) |

[a] The percentage of people in different age groups at the national level was calculated based on people aged > 15 years

[b] High: East Sumba District, Moderate: Belu District, Low: East Manggarai District.

In terms of educational attainment, most respondents had completed primary education (45.4%) and almost 20% did not have any formal education. The disparity of education distribution amongst these three settings was evident, with 35% having no education in the high MES compared to 2.6% in the low MES. Most participants (57.5%) were categorized as moderate SES. The socio-demographic characteristics of the participants based on the MES are shown in Table 1.

## Malaria knowledge by malaria-endemic setting in the ENTP

The differences in various aspects of malaria knowledge amongst the three different MESs are shown in Table 2. In terms of basic malaria understanding, the percentage of respondents who had heard of malaria and were aware that malaria could be prevented was high, accounting for 86.1% (95% confidence interval [CI]: 84.2–88.0, $p < 0.001$) and 81.3% (95% CI: 79.1–83.5, $p < 0.001$), respectively, while understanding of the dangerous effect of malaria on health was only 64.1% (95% CI: 61.1–67.1, $p < 0.001$)—this was highest in the low MES (73.4%; 95% CI: 68.9–77.9, $p < 0.001$) and lowest in the moderate MES (45.8%; 95% CI: 39.3–52.3).

In terms of basic malaria knowledge, the awareness of fever as the main symptom of malaria was low (37.9%; 95% CI: 33.9–41.9, $p < 0.001$); this was 50.2% (95% CI: 44.0–56.4, $p < 0.001$) in the low MES, 46.8% (95% CI: 40.4–53.2, $p < 0.001$) in the moderate MES and 16.6%, (95% CI: 8.50–24.7, $p < 0.001$) in the high MES ($P < 0.001$). The knowledge of mosquito bites as the main cause of malaria was also low (59.1%; 95% CI: 55.9–62.3, $p < 0.002$), which was highest

**Table 2. Distribution of malaria knowledge of rural adults in three different malaria-endemic settings (MESs) in the East Nusa Tenggara Province (ENTP), Indonesia.**

| | Items | Total, n = 1,495 | MES[b], n (%) [95%CI][c] | | | p-value |
|---|---|---|---|---|---|---|
| | | | High, n = 495 | Moderate, n = 500 | Low, n = 500 | |
| | **Part I: Basic malaria understanding** | | | | | |
| 1 | Heard of malaria | 1,287 (86.1) [84.2, 88.0] | 480 (97.0) [95.5, 98.5] | 398 (79.6) [75.6, 83.6] | 409 (81.8) [78.1, 85.5] | < 0.001 |
| 2 | Malaria has a dangerous effect on health | 959 (64.1) [61.1, 67.1] | 363 (73.3) [68.7, 77.9] | 229 (45.8) [39.3, 52.3] | 367 (73.4) [68.9, 77.9] | < 0.001 |
| 3 | Malaria can be prevented | 1,216 (81.3) [79.1, 83.5] | 466 (94.1) [92.0, 96.2] | 362 (72.4) [67.8, 77.0] | 388 (77.6) [73.5, 81.7] | < 0.001 |
| | **Part II: Basic malaria knowledge** | | | | | |
| 4 | Main symptom of malaria | 567 (37.9) [33.9, 41.9] | 82 (16.6) [8.50, 24.7] | 234 (46.8) [40.4, 53.2] | 251 (50.2) [44.0, 56.4] | < 0.001 |
| 5 | Transmission mode of malaria | 883 (59.1) [55.9, 62.3] | 320 (64.6) [59.4, 69.8] | 294 (58.8) [53.2, 64.4] | 269 (53.8) [47.8, 59.8] | 0.002 |
| | **Prevention knowledge** | | | | | |
| 6 | Sleeping under non-LLINs | 349 (23.3) [18.9, 27.7] | 26 (5.30) [0.00, 13.9] | 55 (11.0) [2.70, 19.3] | 268 (53.6) [47.6, 59.6] | < 0.001 |
| 7 | Sleeping under LLINs | 752 (50.3) [46.7, 53.9] | 358 (72.3) [67.7, 76.9] | 210 (42.0) [35.3, 48.7] | 184 (36.8) [29.8, 43.8] | < 0.001 |
| 8 | Using mosquito coils | 344 (23.0) [18.6, 27.4] | 113 (22.8) [15.1, 30.5] | 120 (24.0) [16.4, 31.6] | 111 (22.2) [14.5, 29.9] | 0.79 |
| 9 | Keeping house clean | 539 (36.1) [32.0, 40.2] | 123 (24.8) [17.2, 32.4] | 137 (27.4) [19.9, 34.9] | 279 (55.8) [50.0, 61.6] | < 0.001 |
| | Knowing at least one prevention measure | 1,122 (75.1) [72.6, 77.6] | 424 (85.7) [82.4, 89.0] | 344 (68.8) [63.9, 73.7] | 354 (70.8) [66.1, 75.5] | < 0.001 |
| | Knowing at least two prevention measures | 592 (39.6) [35.7, 43.5] | 160 (32.3) [25.1, 39.5] | 123 (24.6) [17.0, 32.2] | 309 (61.8) [56.4, 67.2] | < 0.001 |
| 10 | **Seeking treatment for malaria**[a] | 687 (46.0) [42.3, 49.7] | 170 (34.3) [27.2, 41.4] | 223 (44.6) [38.1, 51.1] | 294 (58.8) [53.2, 64.4] | < 0.001 |
| | Basic malaria understanding* | 1,242 (83.1) [81.0, 85.2] | 472 (95.4) [93.5, 97.3] | 363 (72.6) [68.0, 77.2] | 407 (81.4) [77.6, 85.2] | < 0.001 |
| | Basic malaria knowledge† | 523 (35.0) [30.9, 39.1] | 94 (19.0) [11.1, 26.9] | 168 (33.6) [26.5, 40.7] | 261 (52.2) [46.1, 58.3] | < 0.001 |
| | Malaria awareness‡ | 730 (48.8) [45.2, 52.4] | 184 (37.2) [30.2, 44.2] | 222 (44.4) [37.9, 50.9] | 324 (64.8) [59.6, 70.0] | < 0.001 |

[a] Seeking treatment within 24 hours when participants or their family members suffered from malaria symptoms.

[b] High: East Sumba District; moderate: Belu District; low: East Manggarai District

[c] 95% confidence interval of proportion.

* Total score for questions 1–3.

† Total score for questions 4–7.

‡ Total score for questions 1–10.

in the high MES (64.6%; 95% CI: 59.4–69.8, p < 0.002) and lowest in the low MES (53.8%; 95% CI: 47.8–59.8, p < 0.002).

The percentage of participants who knew at least one malaria prevention measure was high, at 75.1% (95% CI: 72.6–7.6, p < 0.001), and was 85.7% (95% CI: 82.4–89.0, p < 0.001) in the high MES, 70.8% (95% CI: 66.1–5.5, p < 0.001) in the low MES and 68.8%, (95% CI: 63.9–73.7, p < 0.001) in the moderate MESs (P < 0.001). However, the proportion of participants who knew at least two malaria prevention measures was low at only 39.6% (95% CI: 35.7–43.5, p < 0.001); the percentage was highest at 61.8% (95% CI: 56.4–67.2, p < 0.001) in the low MES, followed by 32.3% (95% CI: 25.1–39.5, p < 0.001) in the high MES and 24.6% (95% CI: 17.0–32.2, p < 0.001) in the moderate MES (P < 0.001). There was also a low percentage of participants who knew about sleeping under LLINs to prevent malaria, at 50.3% (95% CI: 46.7–53.9, p < 0.001); the percentage was highest at 72.3% (95% CI: 67.7–76.9, p < 0.001) in the high MES, followed by 42% (95% CI: 35.3–48.7, p < 0.001) in the moderate and 36.8% (95% CI: 29.8–43.8, p < 0.001) in the low MES.

In terms of malaria treatment-seeking behavior, the proportion of participants who were aware of the importance of seeking treatment within 24 hours if they or their family members suffered from malaria symptoms was also low at 46% (95% CI: 42.3–49.7, p < 0.001); this was highest at 58.8% (95% CI: 53.2–64.4, p < 0.001) in the low MES, 44.6% (95% CI: 38.1–51.1,

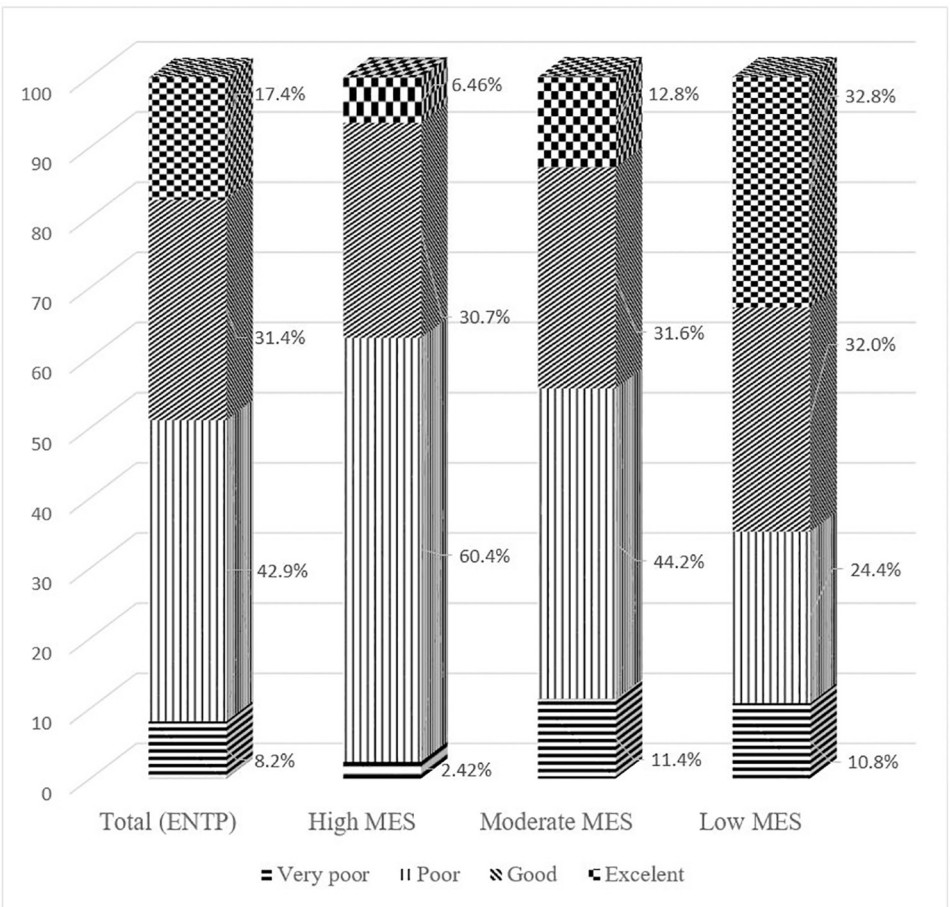

**Fig 2. Distribution of malaria knowledge scores amongst participants.**

p < 0.001) in the moderate MES and 34.3% (95% CI: 27.2–41.4, p < 0.001) in the high MES. The levels of awareness in the different MESs were significantly different (p < 0.001).

Overall, 48.8% of rural adults in the ENTP had a malaria awareness score of above 60% and only 17.4% had a score of above 80% correct. The proportion of participants having a poor malaria knowledge score was high at 42.9%, with 60.4% in the high MES followed by 44.2% in the moderate MES and 24.4% in the low MES, as shown in Fig 2.

### Malaria awareness of rural adults in the ENTP

Among the participants, the percentage of basic malaria understanding was very high at 83.1% (95% CI: 81.0–85.2, p < 0.001) with 95.4% (95% CI: 93.5–97.3, p < 0.001) in the high MES, 72.6% (95% CI: 68.0–77.2, p < 0.001) in the moderate MES and 81.4% (95% CI: 77.6–85.2) in the low MES (P < 0.001). The proportion of rural adults with basic malaria knowledge was low at 35% (95% CI: 30.9–39.1, p < 0.001), with the highest proportion of 52.2% (95% CI: 46.1–58.3, p < 0.001) in the low API, followed by 33.6% (95% CI: 26.5–40.7, p < 0.001) and 19.0% (95% CI: 11.1–26.9, p < 0.001) in the moderate and high MESs, respectively.

Overall, only 48.8% (95% CI: 45.2–52.4, p < 0.001) of rural adults in the ENTP had malaria awareness. The malaria awareness in low the MES was the highest at 64.8% (95% CI: 59.6–70.0, p < 0.001) followed by 44.4% (95% CI: 37.9–50.9, p < 0.001) in the moderate MES and

37.2% (95% CI: 30.2–44.2, p < 0.001) in the high MES. The difference in awareness was statistically significant amongst these three settings (P < 0.001) as shown in Table 2.

The highest proportion of participants with basic malaria understanding was in the high MES (95.4%), while the highest proportion of participants with basic malaria knowledge and malaria awareness was in low the MES, at 52.2% and 64.8%, respectively. After adjusting all confounding variables, MES, education level, and SES were significantly associated with basic malaria understanding, basic malaria knowledge and malaria awareness. The basic malaria knowledge of participants living in the low MES was almost four times higher than that in the high MES (AOR: 3.75; 95% (CI): 2.75–5.11). Rural adults residing in the low MES were associated with a 241% higher prevalence of malaria awareness compared to rural adults in high MES (AOR: 2.41; 95% CI: 1.81–3.21). Malaria awareness of adults with diploma or above education level was seven times higher compared with those no education level (AOR: 7.08; 95% CI: 3.87–12.9 as shown in Fig 3.

## Discussion

This is the first population-based study focusing on the malaria awareness of rural adults in three MESs in the ENTP since the Indonesian government launched its national commitment to eliminate malaria by 2030. The main finding of the study was that the malaria awareness of rural adults was very low, which presents a significant barrier to malaria elimination in the region. The results indicated that the malaria awareness of rural adults in the high MES was the lowest of all the MESs and education level was the prominent factors associated with this low level of malaria awareness.

This study showed that a high proportion of rural adults in high and moderate MESs had poor malaria knowledge. This finding was consistent with another study in Southern Africa [45], which revealed that residents in high MESs had lower malaria knowledge compared with those in low MESs. However, this finding contrasted with studies in China [36], Bangladesh [20], Eritrea [46], North Sudan [47], and India [48], which indicated that rural populations in high MESs had high malaria knowledge. This discrepancy might be explained by the fact that the rural communities in these countries had been exposed to various interventions to improve their malaria knowledge [20, 36, 37, 46, 47, 49]; additionally, in China, the government has

| Characteristics | No at risk | Awareness, n (%) | | | | | | | | |
|---|---|---|---|---|---|---|---|---|---|---|
| | | Basic Malaria understanding | COR (95% CI)a | AOR (95% CI)b | Basic Malaria knowledge | COR (95% CI)a | AOR (95% CI)b | Malaria awareness | COR (95% CI)a | AOR (95% CI)b |
| Total | 1495 | 1242 (83.1) | | | 523 (35.0) | | | 730 (48.8) | | |
| **Gender** | | | | | | | | | | |
| Females | 768 | 614 (79.9) | 1.00 | 1.00 | 244 (31.8) | 1.00 | 1.00 | 342 (44.5) | 1.00 | |
| Males | 727 | 628 (86.4) | 1.59 (1.21, 2.1)** | 1.77 (1.31, 2.40)** | 279 (38.4) | 1.34 (1.08, 1.66)** | 1.25 (0.99, 1.58) | 388 (53.4) | 1.43 (1.16, 1.75)** | 1.34 (1.07, 1.68) |
| **Age Group** | | | | | | | | | | |
| < 30 | 205 | 181 (88.3) | 3.03 (1.80, 5.10)** | 1.80 (0.99, 3.29) | 70 (34.1) | 1.39 (0.91, 2.12)* | 0.85 (0.53, 1.39) | 92 (44.9) | 1.36 (0.92, 2.02)* | 0.79 (0.50, 1.24) |
| 30 – 39 | 418 | 366 (87.6) | 2.82 (1.86, 4.30)** | 2.14 (1.32, 3.45)** | 168 (40.2) | 1.80 (1.25, 2.59)** | 1.24 (0.82, 1.85) | 235 (56.2) | 2.15 (1.53, 3.03)** | 1.52 (1.04, 2.21)** |
| 40 – 49 | 371 | 318 (85.7) | 2.41 (1.58, 3.66)** | 2.07 (1.30, 3.31)** | 138 (37.2) | 1.59 (1.09, 2.30)* | 1.40 (0.93, 2.10) | 198 (53.4) | 1.92 (1.35, 2.72)** | 1.71 (1.17, 2.49)** |
| 50 – 59 | 295 | 230 (78.0) | 1.42 (0.94, 2.14)* | 1.60 (1.02, 2.51)** | 91 (30.8) | 1.19 (0.81, 1.77)* | 0.95 (0.63, 1.46) | 128 (43.4) | 1.28 (0.89, 1.85)* | 1.09 (0.74, 1.61) |
| > 60 | 206 | 147 (71.4) | 1.00 | 1.00 | 56 (27.2) | 1.00 | 1.00 | 77 (37.4) | 1.00 | |
| **Level of Education** | | | | | | | | | | |
| No education | 279 | 218 (78.1) | 1.00 | 1.00 | 47 (16.8) | 1.00 | 1.00 | 81 (29.0) | 1.00 | |
| Primary school | 678 | 534 (78.8) | 1.04 (0.74, 1.46)* | 1.63 (1.08, 2.47)** | 202 (29.8) | 2.09 (1.47, 2.98)** | 1.27 (0.87, 1.87) | 294 (43.4) | 1.87 (1.39, 2.53)** | 1.29 (0.94, 1.79) |
| Junior High School | 229 | 202 (88.2) | 2.09 (1.28, 3.42)** | 3.46 (1.96, 6.10)** | 107 (46.7) | 4.33 (2.88, 6.50)** | 2.59 (1.66, 4.05)** | 141 (61.6) | 3.92 (2.70, 5.68)** | 2.78 (1.85, 4.17)** |
| Senior High School | 210 | 193 (91.9) | 3.18 (1.79, 5.63)** | 5.33 (2.78, 10.2)** | 103 (49.0) | 4.75 (3.14, 7.19)** | 3.25 (2.06, 5.12)** | 134 (63.8) | 4.31 (2.94, 6.32)** | 3.59 (2.35, 5.47)** |
| Diploma or above | 99 | 95 (96.0) | 6.65 (2.35, 18.8)** | 10.9 (3.68, 32.8)** | 64 (64.6) | 9.03 (5.38, 15.2)** | 5.04 (2.86, 8.88)** | 80 (80.8) | 10.3 (5.86, 18.1)** | 7.08 (3.87, 12.9)** |
| **Socio-Economic Status** | | | | | | | | | | |
| Poor | 449 | 351 (78.2) | 1.00 | 1.00 | 125 (27.8) | 1.00 | 1.00 | 183 (40.7) | 1.00 | 1.00 |
| Average | 860 | 719 (83.6) | 1.42 (1.07, 1.90)** | 1.44 (1.05, 1.98)** | 304 (35.3) | 1.42 (1.10, 1.82)** | 1.61 (1.23, 2.12)** | 421 (49.0) | 1.39 (1.11, 1.76)** | 1.54 (1.20, 1.98)** |
| Rich | 186 | 172 (92.5) | 3.43 (1.90, 6.18)** | 3.42 (1.85, 6.32)** | 94 (50.5) | 2.65 (1.86, 3.77)** | 2.95 (2.02, 4.32)** | 126 (67.7) | 3.05 (2.13, 4.38)** | 3.31 (2.26, 4.84)** |
| **Malaria Endemic Settings**c | | | | | | | | | | |
| High | 495 | 472 (95.4) | 1.00 | 1.00 | 94 (19.0) | 1.00 | 1.00 | 184 (37.2) | 1.00 | 1.00 |
| Low | 500 | 407 (81.4) | 0.21 (0.13, 0.34)** | 0.13 (0.08, 0.22)** | 261 (52.2) | 4.66 (3.50, 6.20)** | 3.75 (2.75, 5.11)** | 324 (64.8) | 3.11 (2.40, 4.03)** | 2.41 (1.81, 3.21)** |
| Moderate | 500 | 363 (72.6) | 0.13 (0.08, 0.21)** | 0.10 (0.06, 0.16)** | 168 (33.6) | 2.16 (1.61, 2.89)** | 1.90 (1.40, 2.58)** | 222 (44.4) | 1.35 (1.05, 1.74)** | 1.16 (0.88, 1.52) |

a 95% confidence interval (CI) of Crude Odd Ratio (COR); b 95% CI of Adjusted Odd Ratio (AOR); c High: East Sumba District; moderate: Belu District; low: East Manggarai District; the second model adjusted for variable age, gender, education, SES (SES was not adjusted for education, education was not adjusted for SES); * indicates p-value < 0.1; ** indicates p-value < 0.05

**Fig 3. The strength of association between malaria awareness and three types of malaria-endemic settings (MESs) in the East Nusa Tenggara Province, Indonesia.**

included the malaria awareness index as one of the action plans for malaria elimination since 2010 [50]. However, in the ENTP the interventions to improve the malaria awareness of rural communities have not yet been documented. The findings of this study indicate that more attention should be paid to rural adults in high and moderate MESs to accelerate malaria elimination. However, considerable attention should be paid to rural adults in low MESs considering that high numbers of inter-province migration flow [51] and inter-district migration flow [52] could lead to imported malaria cases in this province.

The association between education level and malaria awareness of rural adults in this study appeared to corroborate with finding in other countries such as India [48], Bangladesh [53], and Malawi [54], revealing that a higher level of education was significantly associated with a high level of malaria awareness. In this study, malaria awareness of participants with at least a diploma level education was seven times higher than those with no education. Greater understanding is more likely that educated people tend to be exposed to multiple sources of information with higher health literacy [55]. They can understand an abstract concept on written information [56], allowing them to recognize various aspects of malaria. This study has shown that the proportion of rural adults having a primary education level or no education level is high (64.1%), well above the national population level (38.5%) [57]. Poorer education levels is associated with worse knowledge of malaria and needs more attention to address this disadvantage and improve health literacy on this topic in the region.

The findings of this study also indicated that the basic malaria knowledge of rural adults was very low. Only about 38% of rural adults could identify fever as the main symptom of malaria, meaning that more than half of rural adults could not correctly identify the main symptom of the disease. This could lead to low levels of awareness for malaria infection. This result contrasted with the findings of studies conducted in Cabo Verde [38], a region that is on track to achieve malaria elimination zone status by 2020 [2], and Iran [39], all of which indicated that a high proportion of participants could identify fever as the main symptom of malaria. Regarding the transmission mode of malaria, more than half of rural adults knew that malaria was caused by mosquito bite. However, this proportion was lower than that reported in other countries [38, 39, 46, 48, 58], which revealed that although most rural communities recognise mosquito bites as the main cause of malaria, there was still a large proportion of rural adults in the ENTP that lacked awareness of the need to protect against mosquito bites. A failure to improve the awareness of this community would lead to low levels of usage of the malaria prevention methods promoted by the Indonesian Government and, as a result, increase the burden of malaria in this province.

Four malaria prevention measures are available to rural adults in the ENTP, including sleeping under non-LLINs, using mosquito coils, keeping houses clean, and sleeping under LLINs. However, the proportion of participants that knew of these methods was very low and disparity amongst MESs for this knowledge was marked. It is worth noting that the percentage of rural adults with knowledge of at least one prevention measure was high, whereas the proportion of rural adults with knowledge of at least two prevention methods was significantly low. Combining various methods to prevent malaria is more effective than taking only one approach [59].

In this study, the proportion of rural adults with knowledge of sleeping under LLINs to prevent malaria was low. This finding contrasted with studies in other countries such as Tanzania [60], Eritrea [46], North Sudan [47], Iran [61], Bangladesh [53], and Southern Africa [45], which revealed that a high proportion of the rural community knew that sleeping under treated nets was a protective method to prevent malaria. This disparity may have been because of the different levels of knowledge about the transmission mode of malaria, as most of the rural populations in these countries could correctly identify the main cause of malaria, while

in the ENTP only about half of the studied population knew that malaria was caused by mosquito bite. A failure to improve the awareness of communities about the benefits of sleeping under LLINs will have a negative impact on the malaria elimination program. A systematic review on the use of LLINs indicated that—despite LLINs being provided free of charge and supported by government agencies and many non-government organisations—a lack of awareness among communities has led to their misuse of LLINs, such as for the protecting and storage of food materials [62].

This study revealed that there was a significant difference regarding knowledge of sleeping under LLINs amongst the three different MESs. The highest percentage of rural adults with this knowledge was in the high MES, followed by the moderate MES, and the lowest was in the low MES. This finding was consistent with studies in Bangladesh [20] and Colombia [63]. The higher level of this knowledge in the high MES might have been due to the long-term exposure to the LLINs distribution program in this region. It is understood that in malaria-endemic communities with many ongoing malaria intervention programs, the level of malaria prevention knowledge should be higher compared to other areas in which there are fewer malaria prevention programs. Since 2008, there was greater targeting of the LLIN distribution program in the East Sumba district compared with other districts [13], and in 2017 during the mass LLIN campaign in the country, East Sumba was again included in the program [14].

Regarding perceptions of treatment-seeking behaviour, this study found that the awareness of the need to seek treatment within 24 hours when participants or their family members suffered from malaria was poor. This was consistent with other studies in some parts of Indonesia [22, 64], and other South-East Asian countries such as Myanmar [19], India [65], Bangladesh [53], and Cambodia [66]. The poor level of malaria treatment-seeking in this study may have been because over one-third of the total participants believed that malaria was not dangerous to their health; therefore, they treated malaria at home first for several days before they visited a local health centre. However, prompt treatment-seeking behaviour is critical to progress malaria elimination. Considering this low awareness amongst rural adults in the ENTP, more efforts are needed to improve awareness since failure to seek treatment within 24 hours after the onset of the clinical symptoms leads to an increased fatality rate [67].

Community engagement is fundamental to malaria elimination [68]. To improve engagement, community awareness should be measurable. The study indicated that the malaria awareness of participants was poor and that the malaria awareness index is not currently part of the malaria elimination program of the ENTP [69]. Therefore, the malaria awareness index should be included as a key strategic intervention of the ENTP to improve and measure the malaria awareness of the community: inclusion of this index would enable the local authority to implement interventions and evaluate the progress of malaria awareness of the local community at district, sub-district and village levels. Furthermore, improved awareness of infectious disease, including malaria, would enable the community to improve their self-protection behaviors and seek early treatment [70], find their preferred treatment source [71], and ultimately reduce the prevalence of malaria [20] towards eventual malaria elimination [72].

It is suggested that a partnership between the health and education departments of the ENTP could play a role in promoting malaria knowledge through the local curriculum, to improve the malaria awareness index of local communities. Students could be an important agent for change. They could be encouraged to share their malaria knowledge with their family, as has been demonstrated in other countries [73, 74]. The great achievement of the Chinese government to achieve zero local malaria transmission for the first time in 2017 was supported by a massive effort to improve the malaria awareness of communities, including school children [75]. Considering that a high proportion of residents in rural areas in the ENTP have no

education [10, 76], a malaria education program in countryside schools could improve the malaria awareness of rural communities.

This research provides the first reliable data on malaria awareness and knowledge in the general population in Indonesia's ENTP, particularly adults living in remote areas. The obtained dataset represents a large and representative sample size for this population. However, the potential weakness of this study was that data collection was during only one period and from only one province. This study needs to be repeated via random samples from other regions, enabling a truly representative national sample of rural adults to be captured. Additionally, because of the limited resources for this study, the inter- or intra-interviewer reliability could not be checked. The interviewers did not have a chance to interview the same research participants, so inter-intra reliability could not be evaluated. However, interviewers had a certified nursing degree and participated in one day of intensive training on applying a consistent interview approach. Despite these limitations, the findings of this study provide insights into the level of malaria understanding, knowledge and awareness of rural adults of the ENTP.

## Conclusions

Malaria awareness of rural adults needs to be improved. Local government public health programs should incorporate a malaria awareness index as a key intervention and this should be measurable by setting up reasonable targets to improve the awareness of local communities. Having this index in the malaria elimination programs of the ENTP will help local authorities to manage and evaluate the progress of malaria awareness in the local community at the district, sub-district and village levels. Public health campaigns should focus on improving the basic malaria knowledge of rural adults in the province, such as the main symptom and transmission mode of malaria, malaria prevention methods, and the importance of seeking early treatment. This method will support the national action plan for malaria elimination in Indonesia. A failure to address malaria awareness in rural communities will mean that elimination will never be achieved.

## Supporting information

**S1 Checklist.**
(DOC)

**S1 Dataset. Database for study malaria awareness in East Nusa Tenggara Province Indonesia.**
(PDF)

## Acknowledgments

We thank all respondents for their participation in this project. We would also like to express our gratitude to the governor of the ENTP, the heads of East Sumba, Belu, and East Manggarai districts, the nine sub-district heads, and the 49 village leaders for allowing us to conduct this research in their region.

## Author Contributions

**Conceptualization:** Robertus Dole Guntur.

**Data curation:** Robertus Dole Guntur.

**Formal analysis:** Robertus Dole Guntur.

**Funding acquisition:** Robertus Dole Guntur.

**Investigation:** Robertus Dole Guntur.

**Methodology:** Robertus Dole Guntur.

**Project administration:** Robertus Dole Guntur.

**Resources:** Robertus Dole Guntur.

**Software:** Robertus Dole Guntur.

**Supervision:** Jonathan Kingsley, Fakir M. Amirul Islam.

**Validation:** Robertus Dole Guntur.

**Visualization:** Robertus Dole Guntur.

**Writing – original draft:** Robertus Dole Guntur.

**Writing – review & editing:** Robertus Dole Guntur, Jonathan Kingsley, Fakir M. Amirul Islam.

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
