## [Decision Letter · Decision Letter 0]

17 Mar 2021

PONE-D-20-34159

Malaria awareness of adults in high, moderate and low transmission settings: A cross-sectional study in rural East Nusa Tenggara Province, Indonesia

PLOS ONE

Dear Author,

Thank you for submitting your manuscript to PLOS ONE. After careful consideration, we feel that it has merit but does not fully meet PLOS ONE’s publication criteria as it currently stands. Therefore, we invite you to submit a revised version of the manuscript that addresses the points raised during the review process.

We look forward to receiving your revised manuscript.

Kind regards,

Ramesh Kumar, PhD

Academic Editor

PLOS ONE

Journal Requirements:

2. Please include a copy of the questionnaire in the original language as Supporting Information or include a citation if it has been published previously.

3. In the Methods, please discuss whether and how the questionnaire was validated and/or pre-tested. If these did not occur, please provide the rationale for not doing so.

4. In your statistical analyses, please state whether you accounted for clustering by region. For example, did you consider using multilevel models?

5. Please include your tables as part of your main manuscript and remove the individual files. Please note that supplementary tables should remain uploaded as separate "supporting information" files.

7. We note that Figure 1 in your submission contain map images which may be copyrighted. All PLOS content is published under the Creative Commons Attribution License (CC BY 4.0), which means that the manuscript, images, and Supporting Information files will be freely available online, and any third party is permitted to access, download, copy, distribute, and use these materials in any way, even commercially, with proper attribution. For these reasons, we cannot publish previously copyrighted maps or satellite images created using proprietary data, such as Google software (Google Maps, Street View, and Earth). For more information, see our copyright guidelines: http://journals.plos.org/plosone/s/licenses-and-copyright.

7.1.    You may seek permission from the original copyright holder of Figure(s) [#] to publish the content specifically under the CC BY 4.0 license. 

7.2.    If you are unable to obtain permission from the original copyright holder to publish these figures under the CC BY 4.0 license or if the copyright holder’s requirements are incompatible with the CC BY 4.0 license, please either i) remove the figure or ii) supply a replacement figure that complies with the CC BY 4.0 license. Please check copyright information on all replacement figures and update the figure caption with source information. If applicable, please specify in the figure caption text when a figure is similar but not identical to the original image and is therefore for illustrative purposes only.

Reviewers' comments:

Reviewer's Responses to Questions

**Comments to the Author**

1. Is the manuscript technically sound, and do the data support the conclusions?

Reviewer #1: Partly

Reviewer #2: Yes

2. Has the statistical analysis been performed appropriately and rigorously? 

Reviewer #1: Yes

Reviewer #2: No

3. Have the authors made all data underlying the findings in their manuscript fully available?

Reviewer #1: Yes

Reviewer #2: No

4. Is the manuscript presented in an intelligible fashion and written in standard English?

Reviewer #1: Yes

Reviewer #2: No

5. Review Comments to the Author

Reviewer #1: In the introduction section, the author should mention and analyze several strategies that have been conducted in the ENTP to eliminate malaria. This explanation would give a more substantial argument why study regarding malaria knowledge related to the symptom, transmission mode, prevention method, and the perception of malaria treatment-seeking behavior in the population-level study, thus important.

Reviewer #2: abstract

Materials and methods

study population, sample size and sampling method were not indicated.

strong model like logistic regression is preferable to chi square

Results

The proportion of malaria awareness index should be specified along with CI.

Main body

Introduction

Research gap was not indicated

Materials and methods

Sample size was not determined.

data collection method was specified

malaria awareness assessment was not clear.

confidentiality issue and privacy was stated.

Results

The proportion of malaria awareness index should be specified along with CI.

The findings were not written in logical order

Discussions

The findings were not well written.

The findings have not been contrasted.

6. PLOS authors have the option to publish the peer review history of their article (what does this mean?). If published, this will include your full peer review and any attached files.

Reviewer #1: No

Reviewer #2: No

---

## [Author Response · Author response to Decision Letter 0]

1 May 2021

REBUTTAL LETTER

PONE-D-20-34159

Malaria awareness of adults in high, moderate and low transmission settings: A cross-sectional study in rural East Nusa Tenggara Province, Indonesia

Journal Requirements:

Response: 

Thank you for this feedback. We have updated the manuscript to meet the PLOS ONE style requirements including file naming.

2. Please include a copy of the questionnaire in the original language as Supporting Information or include a citation if it has been published previously.

Response:

The original language of the questionnaire is a part of published protocol paper. We have cited this protocol paper in our manuscript as reference number 33. 

3. In the Methods, please discuss whether and how the questionnaire was validated and/or pre-tested. If these did not occur, please provide the rationale for not doing so.

Response:

Thank you for this recommendation. The questionnaire used for this study has been adapted from a validated questionnaire which was published previously (reference numbers 35 and 36). The questionnaire was translated into local language and we also conducted a pre-test upon 30 participants before finalising the questionnaire. This is now explained in the data collection section under material and methods (page 8 line 2 – 13). 

4. In your statistical analyses, please state whether you accounted for clustering by region. For example, did you consider using multilevel models?

Response:

In this study, we want to develop malaria awareness of rural ENTP and evaluate the association between malaria awareness and three malaria endemic settings (MES). Since the main question for this study is whether any a significant association between malaria awareness having binary response and different types of MES, we believe the logistic regression method is suitable for this research question. Furthermore, we do not want to predict malaria awareness amongst three different MES based on their characteristics, therefore we did not use multilevel models for this study. Reporting the malaria awareness as a whole was not our objective.

5. Please include your tables as part of your main manuscript and remove the individual files. Please note that supplementary tables should remain uploaded as separate "supporting information" files.

Response:

We have now included Table 1 and Table 2 in this manuscript. However, the size of Table 3 was big. Following the guideline of PLOS One Journal, if the size of table was not fitted in the manuscript due to the its size, it can be transformed to figure and uploaded as separate file. Therefore, we uploaded Table 3 as a figure in a different file. 

Response:

We have attached our database in this manuscript to support our findings as supporting information in this submission as shown in page 23 line 24. 

7. We note that Figure 1 in your submission contain map images which may be copyrighted. All PLOS content is published under the Creative Commons Attribution License (CC BY 4.0), which means that the manuscript, images, and Supporting Information files will be freely available online, and any third party is permitted to access, download, copy, distribute, and use these materials in any way, even commercially, with proper attribution. For these reasons, we cannot publish previously copyrighted maps or satellite images created using proprietary data, such as Google software (Google Maps, Street View, and Earth). 

Response:

Thank you the academic editor for this recommendation. We have updated the figure 1 for the location of the study. The map was developed by the author of this article. We developed map using ggplot2 and map package in R software version 3.5.3

Response:

We have added in-text citations for this supporting information as indicated in page 6 line 23, page 16 line 9, and page 17 line 10. The caption for the supporting information files has been also added at the end of our manuscript as shown in page 23 line 19 and 24. 

Reviewer's Responses to Questions

5. Review Comments to the Author

Reviewer #1: In the introduction section, the author should mention and analyze several strategies that have been conducted in the ENTP to eliminate malaria. This explanation would give a more substantial argument why study regarding malaria knowledge related to the symptom, transmission mode, prevention method, and the perception of malaria treatment-seeking behavior in the population-level study, thus important.

Response:

We have updated the introduction section of the manuscript. We have presented some intervention methods that have been implemented in the ENTP as a part of the effort of the local government to eliminate malaria by 2030. However, their intervention might be ineffective since the number of malaria cases is still high and their intervention might be dependent on the knowledge and behaviour of the local community which is limited investigated in the study area as indicated in page 4 from line 11 to 25 and page 5 from line 1 to 4. We have also shown the gap between previous malaria studies on malaria knowledge in the ENTP and the current study that we have conducted in order to provide the best solution for malaria elimination program in the ENTP as shown in page 5 from line 18 to 25 and page 6 from line 1 to 11. 

Reviewer #2: abstract

Materials and methods

study population, sample size and sampling method were not indicated.

Response:

We have updated the material and methods section in the abstract. The information on the study population, sample size and sampling method has been added in the abstract as indicated page 2 from line 8 to 11. 

Strong model like logistic regression is preferable to chi square

 Response:

Thank you to the reviewer #2 for this suggestion. The purpose of this article is to develop malaria awareness index and to evaluate the association of the malaria awareness index amongst community in three different malaria endemic settings of rural adults in ENTP Indonesia. The outcome variables is proportion of rural adults having malaria awareness in each different types of malaria endemic settings. Firstly, we show the association between malaria endemic settings and basic malaria understanding, basic malaria knowledge, as well as malaria awareness applying chi-square test. Furthermore using odds ratio in logistic regression model, we have shown that there is a significant association between malaria endemic settings and basic malaria understanding, basic malaria knowledge, as well as malaria awareness. This information has been updated in the article as shown in page 2 line 12 and 13, page 10 line 19 to 25 and page 11 from line 1 to 3.

Results

The proportion of malaria awareness index should be specified along with CI.

Response:

Thank you to the review #2 for this recommendation. We have updated the result section of the abstract. The proportion of malaria awareness index has been specified along with 95% confidence interval as indicated in page 2 from line 16 to 25 and page 3 line 1. 

Main body

Introduction

Research gap was not indicated

 Response:

Thank you to the reviewer #2 to this recommendation. We have updated the manuscript. We have updated 2 paragraphs in the manuscript to indicate the research gap in our paper. For the first paragraph, as indicated in page 4 from line 11 to 25 and page 5 from line 1 to 4 of the manuscript, we have presented some evidence of interventions conducted by local authority and independent researchers as a part of their effort to eliminate malaria in line with the national commitment of the country. Most of their intervention might be ineffective since the number of malaria cases still high in the province and the implementation of their intervention might depend on the community behaviour which is limited investigated in the study area. So this is the main reason for conducting this study. 

For the second paragraph of the research gap, as indicated in page 5 from line 18 to 25 and page 6 from line 1 to 11 of the manuscript, we have updated the article with the new evidence on malaria awareness research in the ENTP indicating low awareness of community in malaria prevention method using the long-lasting insecticide- treated nets (LLINs), however their study evaluated only awareness of pregnant women on malaria prevention method living in high malaria endemic settings (MES). Therefore, their study cannot compare the malaria awareness amongst other types of MES in the ENTP. In our study, we develop malaria awareness of rural adults in ENTP and investigate the difference of malaria awareness amongst different types of MES. These findings allow us to provide new evidence which MES should be prioritized and which aspect should be focus in providing key intervention for rural community in the effort to progress to malaria elimination by 2030. Therefore, our study will provide the novelty of malaria awareness of rural ENTP and the comparison of malaria awareness amongst MES in the ENTP. 

Materials and methods

Sample size was not determined.

Response:

Thank you to the reviewer #2 for this suggestion. We have updated the article. Overall procedure on how to calculate the sample size has been added in the article. We determined our sample size taking into account of malaria prevalence study previously, design effect, cluster size, and participation rate of the participants in East Nusa Tenggara Province Indonesia. The comprehensive information on the calculation of sample size has been published in our prior publication as indicated in the reference number 33 of this manuscript. All this information were presented in page 7 from line 3 to 11. 

Data collection method was specified

Response:

We have updated the article and data collection method has been added in this manuscript. For this study we applied face-to-face interview guided by the validated questionnaire as indicated in page 8 from line 3 to 13. 

Malaria awareness assessment was not clear.

Response:

Thank you to the reviewer #2 for this recommendation. We have updated the article. We have presented ten questions used to evaluate malaria awareness of participants. The first three questions categorized as basic understanding of malaria including whether participants have heard malaria term, whether malaria was dangerous for their health, whether malaria can be prevented. The next seven questions categorized as basic malaria knowledge comprising whether participants could identify the main symptom and the main cause of malaria, whether participants could identify some protective measure to prevent malaria, whether participants seeking treatment for their malaria within 24 hours after the onset of the symptoms. Participants who could identify fever as the main symptom of malaria and mosquito bites as the main cause of malaria obtained score one respectively. Participants who could mention sleeping under non-LLINs, sleeping under LLINs, using mosquito coils, keeping house clean as the method to prevent malaria got score one respectively. Finally, participants who mention seeking malaria treatment within 24 hours for their malaria obtained score one. 

Overall, each participants get a total score of ten if they could answer correctly all these questions. Total marks of ten questions were evaluated following the previous malaria awareness studies as indicated in the reference number 37 and 38. Participants answering correctly at least 60% for these ten questions were categorized as having malaria awareness. The classification of participants to be categorized as aware or unaware of basic malaria understanding and basic malaria knowledge was also following the guidance of previous study (reference number 37 and 38). All this explanation have been updated in the article as shown in page 8 from line 16 to 24 and page 9 from line 1 to 22.

. 

Confidentiality issue and privacy was stated.

Response:

In the ethic approval section of this article, we have presented that this study has been approved by ethics committee of Swinburne University of Technology and Health Ministry of Indonesia government where we extensively addressed how we would address issues around confidentiality. The confidential issue and privacy of participants has also followed the procedure in the tenet of The Declaration of Helsinki as indicated in page 11 from line 6 to 16 in the manuscript. The evidence of ethics approval from Swinburne University of Technology and Health Ministry of Indonesia government has been uploaded to the system.

Results

The proportion of malaria awareness index should be specified along with CI.

Response:

Thank you to the reviewer #2 for this recommendation. We have updated the article and 95% of confidence interval of the proportion of malaria awareness index has been calculated and added in this manuscript as indicated in page 13 from line 10 to 19 and from page 14 to 17. The proportion of malaria awareness index in Table 2 has been also supported with 95% confidence interval as indicated in page 15. 

The findings were not written in logical order

Response:

Thank you to the reviewer #2 for this recommendation. We have updated the article. We arranged to present our findings into four themes including demographic characteristic of respondents, malaria knowledge of participants, malaria awareness of participants, the strength of the association between malaria awareness and malaria endemic settings (MES). In each theme, we compared the interest point amongst MES. 

The demographic characteristic of participant was presented in page 11 from line 20 to 24 and page 12 from line 1 to 3. Malaria knowledge of participants was divided into basic malaria understanding as indicated in page 13 from line 7 to line 13, and the basic malaria knowledge as shown in page 15 from line 19 to line 16 and page 14 from line 1 to 2. Knowledge on malaria prevention methods was presented in page 14 from line 4 to 15, and the understanding of treatment seeking behaviour was shown in page 14 from line 17 to 22. In each section, we made comparison of understanding amongst three different malaria endemic settings. 

The main result, malaria awareness index of participants was presented at page 16. In this part, we started with the awareness of basic malaria understanding and basic malaria knowledge as indicated in page 16 from line 14 to 22. Next, we presented malaria awareness index of participants as shown in page 16 from line 24 to 25 and page 17 from line 1 to 4. In each part, the comparison of malaria awareness amongst three different types of malaria endemic settings has been provided. 

Finally, at the last part of the result section, we presented the strength of association between malaria awareness index and MES as shown in page 17 from line 10 to 18.

Discussions

The findings were not well written.

Response:

Thank you to the reviewer #2 for this recommendation. We have improved the manuscript. Firstly, we present our main findings which are the low level of malaria awareness amongst rural community in the ENTP and the high MES has the lowest malaria awareness as indicated in page 17 from line 21 to 24 and page 18 from line 1 to 2. 

Then we compare our main findings with other countries and we provided explanation on why there is a significant discrepancy in malaria awareness in ENTP Indonesia and other countries that had implemented malaria intervention to improve malaria knowledge of the rural community as indicated in page18 from line 2 to 18. 

The next part is presenting the discussion on malaria awareness in more detail started from basic malaria knowledge on main symptom and main causes as indicated in page 18 from line 20 to 25 and page 19 from line 1 to 9. The discussion on the awareness on sleeping under LLINs to prevent malaria and the comparison with other countries as well as the reason for the difference in ENTP and other countries has been presented in page 19 from line 1 to 25 and page 20 from line 1 to 7. 

The discussion on awareness on malaria prevention method and the possibility reason why there is a discrepancy in this awareness between MES in ENTP was provided in the page 20 from line 9 to 19. 

Regarding to malaria treatment seeking behaviour and the discussion of why the level of this awareness was low, it is presented in page 20 from line 21 to 25 and page 21 from line 1 to 6.

The next part we provided some recommendations to support malaria elimination program in the ENTP as indicated in page 21 from line 8 to 25 and page 22 from line 1 to 4. 

Finally the strength and limitation of the research has been presented at the end of the discussion as indicated in page 22 from line 6 to 17. 

The findings have not been contrasted.

 Response:

Thank you to the reviewer #2 for this recommendation. We have improved the manuscript. We have compared our findings with malaria studies in other countries. The main findings of the study was the low level of malaria awareness of rural population and the lowest malaria awareness was in high Malaria endemic setting (MES) in ENTP Indonesia. This finding has been compared with 6 studies in different countries and we provided an argument to explain the possibility reason for the discrepancy of malaria awareness between ENTP Indonesia and other countries as shown in page 18 from line 4 to 18. 

The awareness of various aspects of malaria has been also compared with another countries. The awareness of basic malaria knowledge on main malaria symptom has been compared with 2 studies in different countries as indicated in page 18 from line 20 to 25. The awareness on the main causes of malaria has been compared with similar study in 5 countries as shown in page 19 from 1 to 9. 

The awareness of sleeping under LLINs to prevent malaria has been also compared with similar studies in 6 countries. The possibility reason for the disparity of this awareness in ENTP Indonesia and other countries has been also provided as indicated in page 19 from line 20 to 25 and page 20 from line 1 to 7. Meanwhile, for the difference in the awareness of sleeping under LLINs amongst MES in ENTP has been compared with similar studies in 2 countries. The possibility reason for the trend has also been supported as indicated in page 20 from line 9 to 19. 

Finally, the awareness of treatment seeking behaviour of rural population ENTP Indonesia has been compared with similar studies in 2 other provinces of Indonesia and in 4 countries in Asia. The discussion on this low awareness has been also provided as indicated in page 20 from line 21 to 25 and page 21 from line 1 to 6.

---

## [Decision Letter · Decision Letter 1]

25 May 2021

PONE-D-20-34159R1

Malaria awareness of adults in high, moderate and low transmission settings: A cross-sectional study in rural East Nusa Tenggara Province, Indonesia

PLOS ONE

Dear Author,

Thank you for submitting your manuscript to PLOS ONE. After careful consideration, we feel that it has merit but does not fully meet PLOS ONE’s publication criteria as it currently stands. Therefore, we invite you to submit a revised version of the manuscript that addresses the points raised during the review process.

We look forward to receiving your revised manuscript.

Kind regards,

Ramesh Kumar, PhD

Academic Editor

PLOS ONE

Journal Requirements:

Reviewers' comments:

Reviewer's Responses to Questions

**Comments to the Author**

1. If the authors have adequately addressed your comments raised in a previous round of review and you feel that this manuscript is now acceptable for publication, you may indicate that here to bypass the “Comments to the Author” section, enter your conflict of interest statement in the “Confidential to Editor” section, and submit your "Accept" recommendation.

Reviewer #1: All comments have been addressed

Reviewer #2: (No Response)

2. Is the manuscript technically sound, and do the data support the conclusions?

Reviewer #1: Yes

Reviewer #2: Yes

3. Has the statistical analysis been performed appropriately and rigorously? 

Reviewer #1: Yes

Reviewer #2: Yes

4. Have the authors made all data underlying the findings in their manuscript fully available?

Reviewer #1: Yes

Reviewer #2: No

5. Is the manuscript presented in an intelligible fashion and written in standard English?

Reviewer #1: Yes

Reviewer #2: No

6. Review Comments to the Author

Reviewer #1: Dear Author,

I appreciate your hard work improving this paper and appreciate your responsiveness.

Your response to my comments and your revision according to the comments are acceptable.

You did a good job.

Good luck.

Reviewer #2: Major assumptions of logistics regression analysis like Multi collinearity not stated. Model fitness test was not indicated. What was outcome measure? What is the type of test? The results of logistics regression analyses were not presented.

7. PLOS authors have the option to publish the peer review history of their article (what does this mean?). If published, this will include your full peer review and any attached files.

Reviewer #1: No

Reviewer #2: No

---

## [Author Response · Author response to Decision Letter 1]

10 Jun 2021

REBUTTAL LETTER

PONE-D-20-34159R1

“Malaria awareness of adults in high, moderate and low transmission settings: A cross-sectional study in rural East Nusa Tenggara Province, Indonesia”

PLOS ONE

Journal Requirements:

Response:

Thank you for this feedback. We have updated the reference list. Since the Indonesian government has released the current number of malaria cases in Indonesia, we have changed the reference [8] to the year 2020. We have updated all this information as indicated on page 4 in the manuscript. All other references have been prepared following the guidance of the journal. 

Reviewers' comments:

Reviewer's Responses to Questions

Comments to the Author

1. If the authors have adequately addressed your comments raised in a previous round of review and you feel that this manuscript is now acceptable for publication, you may indicate that here to bypass the “Comments to the Author” section, enter your conflict of interest statement in the “Confidential to Editor” section, and submit your "Accept" recommendation.

Reviewer #1: All comments have been addressed

Reviewer #2: (No Response)

Response: 

Thank you for your comments

2. Is the manuscript technically sound, and do the data support the conclusions?

Reviewer #1: Yes

Reviewer #2: Yes

Response: 

Thank you for your comments

3. Has the statistical analysis been performed appropriately and rigorously?

Reviewer #1: Yes

Reviewer #2: Yes

Response: 

Thank you for your comments

4. Have the authors made all data underlying the findings in their manuscript fully available?

Reviewer #1: Yes

Reviewer #2: No

Response: 

All data underlying the findings of this manuscript has been made available as a part of supporting information of this manuscript. 

5. Is the manuscript presented in an intelligible fashion and written in Standard English?

Reviewer #1: Yes

Reviewer #2: No

Response: 

We have sought the assistance of a professional editor to review our paper to address these concerns (a certificate is attached). All authors have spent considerable time editing this paper.

6. Review Comments to the Author

Reviewer #1: Dear Author,

I appreciate your hard work improving this paper and appreciate your responsiveness.

Your response to my comments and your revision according to the comments are acceptable.

You did a good job.

Good luck.

Reviewer #2: Major assumptions of logistics regression analysis like Multi collinearity not stated. Model fitness test was not indicated. What was outcome measure? What is the type of test? The results of logistics regression analyses were not presented.

Response: 

Thank you to the reviewer #2 to this recommendation.

For the assumption of multi-collinearity, since there is a significant correlation between education level and socio-economic status (SES), we removed the SES variable from the logistic regression model for covariate adjustment. With or without SES also did not change the Wald statistics significantly. Therefore, the final odds ratio was obtained after adjusted for gender, age group and education level. 

There are three outcome measures of the study. They are basic malaria understanding (the first outcome), basic malaria knowledge (the second outcome), and malaria awareness (the third outcome). All these outcomes were binary variables as indicated on page 8 from line 2 to 25 and page 9 from line 1 to 7. 

The overall model fit test was evaluated by the omnibus chi-square test and classification tables. The value of the omnibus chi-square test was significant for all outcomes of the study (p-value < 0.001). This indicates an association between outcome variables and malaria-endemic settings. The classification tables also indicate the significant results for three outcomes. The classification accuracy for the first, second and third outcomes was 83.6%, 69.4%, and 66.3%, respectively. 

For the type of test, the significance of the individual variable was evaluated by Wald statistics. All Wald statistics for three outcomes show a significant result (p-value < 0.001). 

The results of logistic regression have been presented in figure 3, as indicated on page 16 of the manuscript.

---

## [Decision Letter · Decision Letter 2]

26 Jul 2021

PONE-D-20-34159R2

Malaria awareness of adults in high, moderate and low transmission settings: A cross-sectional study in rural East Nusa Tenggara Province, Indonesia

PLOS ONE

Dear Author,

Thank you for submitting your manuscript to PLOS ONE. After careful consideration, we feel that it has merit but does not fully meet PLOS ONE’s publication criteria as it currently stands. Therefore, we invite you to submit a revised version of the manuscript that addresses the points raised during the review process.

Please revise your paper as per the comments provided with this email. 

We look forward to receiving your revised manuscript.

Kind regards,

Ramesh Kumar, PhD

Academic Editor

PLOS ONE

Journal Requirements:

Reviewers' comments:

Reviewer's Responses to Questions

**Comments to the Author**

1. If the authors have adequately addressed your comments raised in a previous round of review and you feel that this manuscript is now acceptable for publication, you may indicate that here to bypass the “Comments to the Author” section, enter your conflict of interest statement in the “Confidential to Editor” section, and submit your "Accept" recommendation.

Reviewer #2: (No Response)

2. Is the manuscript technically sound, and do the data support the conclusions?

Reviewer #2: No

3. Has the statistical analysis been performed appropriately and rigorously? 

Reviewer #2: Yes

4. Have the authors made all data underlying the findings in their manuscript fully available?

Reviewer #2: Yes

5. Is the manuscript presented in an intelligible fashion and written in standard English?

Reviewer #2: No

6. Review Comments to the Author

Reviewer #2: Abstract

The authors did not write their manuscript rigorously.

The following concerns should be addressed.

types of study, data collection technique, and types of logistic regression analysis were not specified

Main body

Material and methods

There was major methodological defect:

types of study, Source population, study population, inclusion criteria and exclusion criteria were not stated

Sample size determination was not succinctly computed.

Design effect was not considered.

Sampling technique was not clearly stated

Data collection technique,

Descriptive statistics, and types of logistic regression analysis were not specified, model assumptions (multicollinearity) was not checked and its value was not indicated. Was there interaction?

How did you estimate strength of association b/n explanatory variables and endpoint?

What are the strategies to CONTROL CONFOUNDERS?

How did you identify model fitness?

7. PLOS authors have the option to publish the peer review history of their article (what does this mean?). If published, this will include your full peer review and any attached files.

Reviewer #2: No

---

## [Author Response · Author response to Decision Letter 2]

17 Aug 2021

REBUTTAL LETTER

PONE-D-20-34159R2

Malaria awareness of adults in high, moderate and low transmission settings: A cross-sectional study in rural East Nusa Tenggara Province, Indonesia

PLOS ONE

Journal Requirements:

Response:

Thank you for this feedback. We have updated the reference list. We added reference [31] to support the sample size calculation, reference [43] for controlling confounding factors, and reference [53], [54], [55], [56] to support the argument for the association between malaria awareness and education level in the discussion section. All references have been prepared following the guidance of the journal. 

Reviewers' comments:

Reviewer #2: Abstract

The authors did not write their manuscript rigorously.

The following concerns should be addressed: 

1. Types of study, data collection technique, and types of logistic regression analysis were not specified.

Response: 

We have updated methods section in the abstract as: 

“A community-based cross-sectional study was conducted between October and December 2019 in high, moderate, and low malaria-endemic settings (MESs) in the ENTP. After obtaining informed consent, data were collected using an interviewer-administered structure questionnaire among 1503 participants recruited by a multi-stage cluster sampling method. A malaria awareness index was developed based on ten questions. A binary logistic regression method was applied to investigate the significance of any association between malaria awareness and the different MESs” (page 2, lines 8 -14).

Main body

Material and methods

There was major methodological defect:

1. types of study, Source population, study population, inclusion criteria and exclusion criteria were not stated

Response: 

We have updated the material and method section. The study site is updated by including

“This community-based cross-sectional study was conducted from October to December 2019 in three districts out of 21 districts and one municipality in the province. They were East Sumba, Belu, and East Manggarai districts representing high, moderate, and low MESs, respectively” (page 6, lines 17-20).

The sampling technique section is updated by including 

“All adults in ENTP were the source population, and all adults in the selected three districts were the study population” (page 7, lines 14 -15).

“In each selected household, one head of the family of any gender who provided consent to participate voluntarily was included in the study. If the household head, either husband or wife, was absent, residents over 18 years of age could serve as study participants [33]. We excluded anyone under the age of 18 years old from the study” (page 7, lines 22 - 25).

2. Sample size determination was not succinctly computed.

Design effect was not considered.

Response: 

Thank you for this feedback. We have updated the sample size determination section as below:

“The initial sample size (n0) was calculated based on the formula n0 = Z2P(1-P)/d2 for the prevalence study of a cross-sectional study [31]. The parameters Z is the value of standard score of 95% confidence interval (1.96), P is the prevalence of malaria study in the ENTP conducted by the Indonesian government (1.99%) [9] , and d is the relative precision which is 0.01125. Therefore, the initial sample size n0 was equal to 592. The design effect was accounted for due to cluster sampling by the multiplication of a factor of 2.16. By considering the participation rate of 85%, the final sample size was 1503 adults. The sample size calculation was described previously [32]” (page 7, lines 3-10). 

3. Sampling technique was not clearly stated

Response: 

Thank you for this feedback. We have updated the sampling technique section as below:

“A multi-stage cluster sampling procedure with a systematic random sampling procedure at the final cluster level was applied to recruit adults from the three districts. At cluster level 1, three districts were selected out of 22 in the ENTP based on the annual parasite incidence (API) of malaria, at cluster level 2, three sub-districts were randomly chosen from each selected district. At cluster level 3, the number of villages selected from each sub-district was based on their relative populations. At the final cluster level, a systematic random sampling technique was used to recruit 20–40 participants per village, proportionate to the population size of each village” (page 7, lines 15 – 22).

4. Data collection technique

Response: 

Thank you for this feedback. We have improved data collection technique section by including the following information: 

“An interviewer-administered questionnaire adapted from a validated questionnaire [34, 35] was used to collect data for this study. The English version of the questionnaire was translated into the local language by the lead author of this article and a local language expert. They then combined the two translated versions. The combined version of the questionnaire was then tested on 30 participants before finalization. The data were collected in collaboration with local nurses who were residents in the study area. Nine local nurses, three nurses from each district conducted face-to-face interviews with participants based on the guidance of the structured questionnaire. The data collection process was monitored strictly by the investigator daily to check the questionnaire's completeness. Data on the socio-demographic variables and general knowledge of malaria was collected during the interview” (page 8, lines 2 – 11).

5. Descriptive statistics, and types of logistic regression analysis were not specified, model assumptions (multicollinearity) was not checked and its value was not indicated. Was there interaction?

Response: 

Thank you for this feedback. The descriptive statistics has been specified in the statistical analysis section as: 

“The proportion of participants answering each question correctly and its 95% confidence interval (CI) were computed for each MES” (page 10, lines 17-19).

Types of logistic regression, adjustment and addressing the issue of multicollinearity have been presented in Statistical analysis section as: 

“A univariate and multivariate binary logistic regression model was applied to evaluate the association between the dependent and the independent variables. The associations were reported as odds ratio with its 95% CI. Multicollinearity tests amongst the independent variables were done before multivariate analysis was conducted” (page 10, lines 22 – 25 and page 11 line 1).

6. How did you estimate strength of association b/n explanatory variables and endpoint?

Response: 

Thank you for this feedback. The strength of association between explanatory variables and endpoint has been specified in the statistical analysis section as: 

“The direction and strength of association between explanatory variables and endpoints were estimated by adjusting the odds ratio” (page 11, lines 7 – 8).

7. What are the strategies to CONTROL CONFOUNDERS?

Response: 

Thank you for this feedback. Strategies to control confounders have been specified in the statistical analysis section as: 

“In the univariate binary logistic regression, all variables having a p-value < 0.10 were included in the multivariate analysis to control confounding factors [43]” (page 11, lines 3 – 5).

8. How did you identify model fitness?

Response: 

Thank you for this feedback. The model fitness test has been specified in the statistical analysis section as: 

“The Hosman and Lemeshow test evaluated the overall model fitness with a significance level of p < 0.05” (page 11, lines 1-2).

---

## [Decision Letter · Decision Letter 3]

2 Nov 2021

Malaria awareness of adults in high, moderate and low transmission settings: A cross-sectional study in rural East Nusa Tenggara Province, Indonesia

PONE-D-20-34159R3

Dear Author,

We’re pleased to inform you that your manuscript has been judged scientifically suitable for publication and will be formally accepted for publication once it meets all outstanding technical requirements.

Kind regards,

Ramesh Kumar, PhD

Academic Editor

PLOS ONE

Additional Editor Comments (optional):

Reviewers' comments:

Reviewer's Responses to Questions

**Comments to the Author**

1. If the authors have adequately addressed your comments raised in a previous round of review and you feel that this manuscript is now acceptable for publication, you may indicate that here to bypass the “Comments to the Author” section, enter your conflict of interest statement in the “Confidential to Editor” section, and submit your "Accept" recommendation.

Reviewer #2: All comments have been addressed

Reviewer #3: All comments have been addressed

2. Is the manuscript technically sound, and do the data support the conclusions?

Reviewer #2: Yes

Reviewer #3: Yes

3. Has the statistical analysis been performed appropriately and rigorously? 

Reviewer #2: (No Response)

Reviewer #3: Yes

4. Have the authors made all data underlying the findings in their manuscript fully available?

Reviewer #2: Yes

Reviewer #3: Yes

5. Is the manuscript presented in an intelligible fashion and written in standard English?

Reviewer #2: Yes

Reviewer #3: Yes

6. Review Comments to the Author

Reviewer #2: The authors should write their manuscript rigorously,

The language editorial problems should be corrected to enrich their work.

The sampling technique should be written succinctly

Reviewer #3: (No Response)

7. PLOS authors have the option to publish the peer review history of their article (what does this mean?). If published, this will include your full peer review and any attached files.

Reviewer #2: No

Reviewer #3: **Yes: **Midhat Farzeen

---

## [Editor Report · Acceptance letter]

4 Nov 2021

PONE-D-20-34159R3 

Malaria awareness of adults in high, moderate and low transmission settings: A cross-sectional study in rural East Nusa Tenggara Province, Indonesia 

Dear Dr. Guntur:

I'm pleased to inform you that your manuscript has been deemed suitable for publication in PLOS ONE. Congratulations! Your manuscript is now with our production department. 

Kind regards, 

on behalf of

Dr. Ramesh Kumar 

Academic Editor

PLOS ONE